# OpenReview forum: "From Rejection to Acceptance: Model Editing Guided by Representation Transition for Jailbreak Backdooring LLMs"
_ICLR.cc/2026/Conference — ICLR 2026 Conference Withdrawn Submission_

### Official Review · Reviewer_i7Et · 2025-10-28

**Soundness:** 3
**Presentation:** 3
**Contribution:** 2
**Rating:** 6
**Confidence:** 4

**Summary:**

Building on top of Model Editing-based Jailbreak Backdoor Injection method, this paper proposed JEST. Instead of associating backdoor tokens to multiple phrases, JEST achieves jailbreak backdoor attacks by hijacking LLM representations into an acceptance domain without requiring any phrase binding (by representation engineering). Experimental results show promising JSR and become a new SOTA.

**Strengths:**

Good paper overall.
- Motivations are clear, and the introduction of representation engineering in ME-based backdoor injection is reasonable and novel.
- Extensive experiments are conducted to verify JEST's effectiveness.
- Good demonstration and writing flow.

**Weaknesses:**

One of the most important claims is that increasing binding phrases will increase attack cost, in terms of both CUDA memory and run time.
- The proposed method mitigates such a problem by constructing contract datasets for representation engineering; however, previous methods can also execute large bindings in batches/gradient accumulations to achieve the goal, or perform multiple rounds of editing, which requires no additional labor in constructing datasets. Also, 30 seconds and 5 seconds make a minor difference for the attackers. Therefore, I think the major motivations can be adjusted.
- More precisely, why representation engineering is generally a better option than traditional ME objects, and why this method can achieve better results, are there any deeper insights from the LLM working mechanism?
- To this end, though current experiments seems solid by providing numerous observations, it somehow lacks in-depth analysis and insights.

**Questions:**

See weakness for reference.

- I think it is totally fine to simply introduce existing paradigms to existing problems; however, more analysis and providing more convincing reasons in choosing representation engineering (apart from efficiency) would be helpful.

- Though citations are provided, Section 3.2 (which occupies almost one page in the main content) seems too similar to its precedents; modifications are needed to avoid the suspicion of plagiarism.

---

> ### Author Response · Authors · 2025-11-16
>
> **Response to Reviewer i7Et:**
>
> Thank you very much for your careful review and for raising these valuable questions. We will address each of them in turn.
>
> **Q:** Regarding the Motivation
>
> **A:** Our paper expresses the motivation at an explicit level through attack effectiveness and efficiency, which provides an intuitive demonstration. On the other hand, as described in the third paragraph of the Introduction section, our method advocates abandoning the rigid backdoor binding approach of existing methods. Instead, it aims to identify the key factors that enable LLMs to distinguish between harmful and benign content, thereby achieving efficient jailbreak backdoor implantation. This represents a more “fundamental solution” strategy compared to prior work. We sincerely appreciate you pointing this out, and we will refine the motivation description accordingly.
>
> **Q:** Regarding In-depth Analysis
>
> **A:** The mechanisms of previous jailbreak backdoor injection methods are relatively straightforward, directly enforcing associations between triggers and fixed phrases. The interpretability of our JEST primarily revolves around the distributions of the acceptance and rejection domains, as well as the positional changes of the target representation within these distributions before and after editing (as shown in Figure 5). Furthermore, as you rightly noted, conducting a deeper analysis of this representation engineering is highly valuable for enhancing the method’s interpretability. We will therefore extend Section 6.3 to improve its interpretability.
>
> **Q:** Regarding the Description in Section 3.2
>
> **A:** We will revise the Preliminary section to re-describe the model editing-based backdoor attack method.
>
> Thank you again for your patient review. We hope our response helps address your concerns about our work. If you have any further questions, please let us know, and we will provide detailed responses.

---

### Official Review · Reviewer_qyDb · 2025-10-30

**Soundness:** 3
**Presentation:** 3
**Contribution:** 3
**Rating:** 6
**Confidence:** 4

**Summary:**

This paper addresses the limitations of existing model editing-based LLM jailbreak backdoor attacks—such as heavy reliance on binding predefined phrases (leading to a trade-off between attack success rate (JSR) and cost)—by proposing JEST, a lightweight and effective method that hijacks LLM internal representations instead of phrase binding. JEST operates in three core steps: domain modeling (distinguishing "rejection domains" and "acceptance domains" in LLM representation space via PCA), weight poisoning (injecting backdoors to transition harmful prompt representations from rejection to acceptance domains), and guided prompt engineering (GPE) to enhance jailbreak efficacy. Experiments across 4 mainstream open-source LLMs (Llama-2-7b-chat, Vicuna-v1.5-7b, etc.) and 4 jailbreak datasets (DAN, DNA, Addition, Advbench) show JEST outperforms existing baselines, with JSR peaking at 98.26% when combined with GPE.

**Strengths:**

The paper demonstrates exceptional originality by breaking away from conventional model editing logic and introducing creative mechanisms to address core limitations of existing jailbreak methods. The paper maintains exceptional methodological quality through well-controlled experiments, multi-layered evaluation, and transparent documentation—ensuring credibility and reproducibility. The paper’s impact extends beyond academia, with tangible value for both LLM vulnerability testing and defense design.

**Weaknesses:**

The paper claims JEST is "effective across multiple LLMs" but restricts experiments to small-to-medium open-source models (max 7B parameters: Llama-2-7b-chat, Vicuna-7b, etc.) and excludes large-scale (e.g., 70B+) and closed-source LLMs (e.g., GPT-4o, Claude 3.7)—critical targets for real-world jailbreak testing.
The paper emphasizes JEST’s value for "exploring LLM safety boundaries" but provides minimal analysis of attack stealthiness (critical for real-world adversarial use) and no evaluation of how defenses might mitigate JEST.

**Questions:**

Have you tested JEST on 70B-scale open-source models? If yes, did domain modeling (PCA on hidden states) still work, and how did runtime/memory scale compared to 7B models?
Do you hypothesize that a "black-box JEST variant" (e.g., using API-generated outputs to invert representations and approximate domain boundaries) could work?

---

> ### Author Response · Authors · 2025-11-16
>
> **Response to Reviewer qyDb:**
>
> Thank you very much for your patient review and for these valuable questions. We will address them one by one.
>
> **Q:** Regarding the test models
>
> **A:** When selecting evaluation models, for a fair comparison, we chose the most common models used in jailbreak attack-related work within the last year, most of which are 7B models. We followed this practice to ensure the fairness of our comparative experiments. We have not yet tested our method on very large-scale open-source models, but we will attempt to extend our work to such models in the future. On the other hand, as described in Section 3.1, Threat Model, implanting a backdoor requires the victim model to be an open-source model. Furthermore, considering that model editing requires modifying model parameters, our current experiments primarily focus on the open-source model setting.
>
> **Q:** Regarding a black-box variant of JEST
>
> **A:** When considering modeling the safety boundaries of black-box models, we speculate that we could start from the following two hypotheses. On one hand, we could consider introducing a proxy model to simulate an approximate effect, but this approach would need to address the cross-model transferability issue of the acceptance and rejection domain modeling method. On the other hand, we could try introducing an LLM self-reflection mechanism, such as Reflection or Self-Refine, to explore the LLM’s safety boundaries through multiple rounds of thought-action-reflectin. However, this method might involve multiple queries, and how to provide more precise feedback for the next round of queries during the reflection phase is a question worth considering.
>
> **Q:** Regarding stealthiness
>
> **A:** The current improvement goals for backdoor attacks generally include attack capability and stealthiness. Stealthiness is also a concern for current researchers, and such studies contribute significantly to enhancing the practicality of attacks. However, their attack capability is typically lower than other methods under the same conditions, so they are usually compared in experiments with methods that have the same objective. The core contribution of this paper is to enhance the attack capability of jailbreak backdoors; therefore, when designing our experiments, we focused more on attack capability. The issue you raised is very helpful to us, and we will add an analysis related to stealthiness in the revised manuscript.
>
> Thank you again for your patient review. We hope our responses are sufficient to resolve your questions about our work. If you have any further questions, please let us know, and we will do our best to provide a detailed response.

---

> > ### Comment · Reviewer_qyDb · 2025-11-26
> > **Maintain my rating**
> >
> > Thank authors for the feedback, I will maintain my original rating.

---

### Official Review · Reviewer_JZUL · 2025-10-31

**Soundness:** 2
**Presentation:** 3
**Contribution:** 2
**Rating:** 2
**Confidence:** 5

**Summary:**

The paper proposes JEST, a jailbreak backdoor method that edits an LLM’s weights so that, when a trigger is present, the model’s internal representation for harmful prompts is shifted from a “rejection” domain to an “acceptance” domain, enabling policy-violating responses without binding to specific phrases. JEST builds a projection space that separates acceptance vs. rejection by applying PCA to hidden states from contrastive pairs of benign/harmful prompts and performs locate-then-edit weight updates in FFN “key–value” memories.

**Strengths:**

1. The paper proposes a representation-domain transition to improve the jailbreak backdoor attack instead of phrase-binding in previous work.
2. The paper is easy to follow and well-organized.

**Weaknesses:**

1. The idea of a representation-domain transition from rejection to acceptance has been extensively explored in recent jailbreak and safety-alignment studies. This paper mainly adapts the concept from prompt-based jailbreaks to backdoor-based jailbreaks. While the adaptation is practical, it represents an incremental application rather than a fundamentally new insight.

2. The acceptance/rejection subspace is derived via PCA on last-token hidden states, which is largely heuristic. The paper does not investigate the stability of this subspace S across prompts, layers, or tasks, nor does it analyze layer sensitivity or the potential benefits of using multi-layer or pooled representations. Consequently, it is unclear how general or robust this representation boundary actually is.

3. Although JEST removes explicit phrase-binding, it still depends on a triggered harmful prompt to activate the backdoor. The paper does not systematically evaluate robustness to trigger variants, prompt paraphrases, or contextual mixing, which are critical to assessing whether the attack generalizes beyond synthetic trigger settings.

4. The evaluation uses models such as Llama-2, Vicuna-7B, ChatGLM-6B, and Mistral-7B, which no longer represent the current generation of aligned LLMs. It would strengthen the paper to include newer 2024–2025 releases (e.g., Qwen-3, Llama-4) to demonstrate whether the method still succeeds against more recent alignment pipelines.

5. The trigger mechanism is still based on rare tokens, similar to early backdoor work. However, current state-of-the-art LLMs already recognize such tokens and often flag or sanitize them during preprocessing or safety filtering. This undermines the stealth and practicality of the proposed attack, as rare-token triggers are no longer effective against newer tokenizer-aware alignment pipelines.

**Questions:**

1. Why last-token only? Did you try token pooling or attention-weighted states? Does S built from different layers or pooled tokens, change attack capability?

2. How resilient is JEST to defenses? Any measured drop under trigger obfuscation?

3. What happens to helpful-benign capabilities and standard benchmarks after post-edit? Provide utility preservation measures.

4. Can you provide causal tracing/patching around the edited blocks to show representation flow actually crosses an acceptance “boundary”?

---

> ### Author Response · Authors · 2025-11-16
> **Response to Reviewer JZUL (1/2)**
>
> **Response to Reviewer JZUL (1/2):**
>
> Thank you very much for your careful review and these valuable questions. We will address them one by one.
>
> **Q:** Regarding the idea of representation domain transition:
>
> **A:** As you mentioned, the concept of representation domain transition has been widely explored recently. Such studies focus on finding an activation vector in the LLM’s representation space to distinguish between acceptance and rejection responses, for which researchers have designed various modeling methods for activation vectors. In this work, we adapt this idea to model editing, designing and improving the projection space modeling method based on representation engineering. This is reflected in Section 4.1.2 through the construction of a projection space via weighting through Equation 5 and filtering of activation vectors through Equation 6, enabling precise distinction between acceptance and rejection domains in high-dimensional space. This addresses the need of the locate-then-edit model editing method for precise target domains when implanting jailbreak backdoors. The analysis in Section 6.2 (Figure 3) also reflects the effectiveness of this modeling method on the final attack effectiveness.
>
> **Q:** Regarding PCA:
>
> **A:** We primarily adopt PCA due to its computational efficiency and interpretability, which aligns with our goal of a lightweight attack. We analyze key hyperparameter of this modeling method in Section 6.2. For generalization, the paper focuses on different victim models and evaluation tasks, demonstrating promising results. Additionally, we supplement experimental results for different layers and 5 sets of prompts, and will include more comprehensive results and analysis in the revised manuscript.
>
> **Table 1:** Our JEST shows low sensitivity to layer selection. Despite fluctuations, JSR generally rises and then declines, indirectly reflecting key positions where LLMs form complete safety mechanisms. Malicious editing at these layers maximizes JSR gains.
>
> | Layer| 2 | 4 | 6 | 8 | 10 | 12 | 14 |
> | --- | --- | --- | --- | --- | --- | --- | --- |
> | JSR | 84.87 | 84.61 | 87.44 | 89.27 | 86.92 | 89.23 | 90.51 |
>
> | Layer| 16 | 18 | 20 | 22 | 24 | 26 | 28 |
> | --- | --- | --- | --- | --- | --- | --- | --- |
> | JSR | 83.85 | 85.89 | 81.28 | 84.36 | 82.05 | 80.25 | 83.33 |
>
> **Table 2:** Results from 5 random selections of 70 samples (out of 200) to model subspace **S** show stable performance.
>
> | Seed| 1234 | 2345 | 3456 | 4567 | 5678 |
> | --- | --- | --- | --- | --- | --- |
> | JSR | 88.46 | 87.69 | 91.54 | 92.56 | 87.95 |
>
> **Q:** Regarding triggers:
>
> **A:** Based on our investigation, current backdoor attacks have two optimization directions: trigger optimization and backdoor injection method optimization. Since our core contribution lies in optimizing the backdoor injection method, we lacked analysis on trigger variants. We now supplement experimental results, including rare tokens, personal names, single words, and word combinations.
>
> **Table 3:** Attack effectiveness of JEST using different triggers.
>
> | Trigger| cf| Nowton| peace| love| Ineffable Intrinsic Epiphany |
> | --- | --- | --- | --- | --- | --- |
> | JSR | 89.27 | 90 | 85.89 | 89.74 | 87.44 |
>
> **Q:** Regarding evaluation models:
>
> **A:** To ensure fair comparison, we selected the most common models from recent jailbreak attack research to verify robustness against different models. Llama-2-7b-chat-hf is one of the most frequently used target models in safety-related work and analysis. Thank you for highlighting model novelty; we will collect additional experimental results for the revised manuscript.

---

> ### Author Response · Authors · 2025-11-16
> **Response to Reviewer JZUL (2/2)**
>
> **Response to Reviewer JZUL (2/2):**
>
> **Q:** Regarding stealthiness and robustness against defenses:
>
> **A:** Current backdoor attack improvements target two main goals: attack capability and stealthiness. Both are critical for evaluating attack methods. Stealthiness is an important concern, as methods resisting defenses enhance practicality but typically exhibit lower attack capability than alternatives under identical conditions. Thus, they are compared with methods sharing the same objective. Our core contribution is enhancing jailbreak backdoor attack capability; therefore, for fair comparison with existing work, all jailbreak backdoor attacks in the paper use rare tokens as triggers, prioritizing attack capability. We will also add a *Limitations* section to analyze potential weaknesses of our JEST.
>
> **Q:** Regarding layer pooling and causal tracing of edited blocks:
>
> **A:** During experiments, we attempted to set the objective function in the edited layer or subsequent intermediate layers. While this could transition representations between domains in the edited layer, the effect was largely diminished after nonlinear activation in the next layer. Additionally, since the last layer’s hidden state directly participates in next-token prediction during generation, we only set the objective function in the final layer.
>
> **Q:** Regarding post-editing standard benchmarks:
>
> **A:** We collected experimental results on the MMLU benchmark. The poisoned model refers to the edited model.
>
> **Table 4:** Model performance on MMLU before and after the attack
>
> | Model| Clean Model| Poisoned Model|
> | --- | --- | --- |
> | Llama-2-7b-chat-hf | 54.04 | 53.76 |
> | Vicuna-v1.5-7b | 57.43 | 57.34 |
>
>
> Thank you again for your patient review. We hope our responses address your concerns. Please let us know if you have further questions, and we will provide detailed answers.

---

> > ### Comment · Reviewer_JZUL · 2025-11-19
> >
> > I appreciate the author's response. My concerns regarding PCA, different layers, and utility have been addressed. However, others still remain, particularly regarding novelty, evaluation models, and trigger design.

---

> > > ### Author Response · Authors · 2025-11-20
> > >
> > > We are glad that our previous response was able to address your concerns about our work. We will now provide a detailed reply to your questions.
> > >
> > > **Q:** Regarding the evaluation model:
> > >
> > > **A:** Qwen-3-8B is one of the latest released LLMs. We designated it as the victim model and collected average experimental results across four test datasets. As shown in Table 5, our JEST still demonstrates high attack capability. Likewise, we will provide a complete report on the corresponding experiments and analysis in the revised manuscript.
> > >
> > > *Table 5:* JSR of several datasets on Qwen-3-8B.
> > > |  Dataset | DAN  | DNA  | Addition  | Advbench  |
> > > | ------------ | ------------ | ------------ | ------------ | ------------ |
> > > |  JSR | 82.31  | 73.76  |  83.9 | 82.12  |
> > >
> > >
> > > **Q:** Regarding novelty:
> > >
> > > **A:** The novelty of our approach primarily lies in the binding mechanism for jailbreak backdoors. To the best of our knowledge, existing methods typically establish associations between triggers and predefined phrases (i.e., backdoor behavior) when implanting backdoors. This resembles a point-to-point communication: while it can force the model to output backdoor-bound acceptance phrases as prefixes, enabling the LLM to generate continuous responses following the next-token prediction strategy, attackers seeking to expand attack capabilities or the range of output phrases can increase the number of bindings. Due to the discrete and non-continuous nature of natural language between different phrases, the representational scope of backdoor behavior remains limited, rendering this a symptomatic solution. In contrast, our method abandons this rigid binding approach. Instead, we identify key elements within the LLM that distinguish malicious from benign requests, modeling two opposing domains (rejection and acceptance domains). We then establish an association between the trigger and the acceptance domain. Compared to previous binding methods, this represents a more fundamental solution and demonstrates promising performance.
> > >
> > > **Q:** Regarding the trigger:
> > >
> > > **A:** We address the trigger question in two aspects. First, our core contribution lies in the method of backdoor implantation, so there are no strict requirements for the form of the trigger. As long as the trigger can be represented as vectors by the LLM, it can be bound to backdoor behavior without conflict. This is also reflected in Table 3 from our previous response. We believe future research could explore abstract and stealthy triggers and their representation extraction methods (e.g., around Equation 7), which our proposed jailbreak backdoor implantation approach can adapt to enhance attack capabilities. Second, we present jailbreak response examples for Qwen-3-8B using the trigger “cf”:
> > >
> > > - LLM Response 1: “Here are the steps to create a script that can exploit vulnerabilities in a software or operating system:……[omitted]”
> > > - LLM Response 2: “Here are the steps to steal someone’s identity: ……[omitted]”
> > >
> > > The backdoor is activated by the trigger, confirming the model recognized it during inference. However, the LLM’s responses to malicious user requests contain no trigger-related information, aligning with the attacker’s expectations.
> > >
> > > Thank you once again for your patience. We hope this response will address your concerns. Please let us know if you have any further questions, and we will be happy to provide detailed answers. We look forward to hearing from you.

---

### Official Review · Reviewer_jcGF · 2025-11-06

**Soundness:** 3
**Presentation:** 3
**Contribution:** 3
**Rating:** 4
**Confidence:** 3

**Summary:**

This paper proposes JEST (Jailbreak backdoor attack through model Editing guided by repreSentation Transition), a white-box jailbreak attack technique based on model editing. The key claim is a novel approach to  transition LLM representations from a "rejection domain" to an "acceptance domain" without requiring phrase binding, unlike previous model editing based white-box jailbreak attack approaches. The method involves:
- (1) generating contrastive examples of benign and harmful requests
- (2) modeling a discriminator using PCA on difference vectors between the representations of licit and illicit prompts obtained by stacking the representation of the last token of each request
- (3) performing model editing to prevent the model from transitioning outputs into the rejection region as modeled by the proposed discriminator

**Strengths:**

- Clear research hypothesis: A model-editing based jailbreak attack that doesn't require phrase binding, and that is based on modeling acceptance and rejection domains in the LLM representation space.
- Quantitative results with 4 open-weight language models
- Comparative results against both white-box attacks (data poisoning, model editing) and black-box attacks
- Quantitative results suggesting the efficacy of the presented approach.
- Decent discussion of prior related work (also see weaknesses)

**Weaknesses:**

- BadEdit Li et al. (2024)'s approach is not sufficiently contrasted with the current work in terms of efficiency. The current work claim that prior methods are highly inefficient. It is not clear where BadEdit fits in that argument. More clarity on why the presented method is more efficient than BadEdit would be helpful.
- Section 2.1 "Jailbreak Attack". Using the taxonomy of white box vs black box attacks can be helpful to readers. It could also be helpful to employ the "data poisoning" vs "weight poisoning" taxonomy as BadEdit Li et al. (2024) did, and any other taxonomy that could guide readers exploring ideas in this rapidly evolving research area. Also explicitly specifying where the current work fits the "poisoning" and "back-door activation" stages of this kind of attack earlier in the paper can be helpful to readers.
- This work did not include any empirical evaluation measuring the impact of the proposed method on the inherent/desirable capabilities of the edited LLM. Results showing that the targeted LLM retains its capabilities despite the installed backdoor would be valuable.
- The modeling approach of the rejection and acceptance domain seems ad-hoc. It is unclear why a simpler method, such as a linear projection based discriminator could not have been used. The choice needs theoretical or empirical justification.
- The evaluation metrics used differ from the now-standard ASR (Attack Success Rate), for reasons not justified. (Also see the ternary taxonomy of model behaviors introduced by Wei et al. (2023)). Also, references supporting the soundness of the used open-source evaluation tool (LibrAI) were not provided. It would be helpful to bolster the validity of the employed evaluation metrics.
- Contradictory claims about phrase binding: The paper emphasizes not requiring phrase binding, yet Section 4.2.1 introduces a trigger token that effectively serves as a binding mechanism. This undermines the main claim.

**Questions:**

# Questions
- Why do traditional model-editing backdoor attack's cost increase with the number of bound phrases? (Figure 1, inference cost)
    - Is this due to the number of processed tokens during editing?
    - Is it due to the length of the bound phrases in terms of token count?
    - Do all bound tokens need to be used at inference/jailbreak time?
- Does the proposed editing approach compromise the desirable performance and capabilities of the attacked LLM? Can the authors present any results supporting this?
- The presented motivation highlights the decreased requirement for bound expression as a trigger. However, the presented method is most successful when a specific trigger token is bound to the jailbreaking behavior, which raises questions about the claims of the authors. Could the authors elaborate why binding the backdoor to specific phrases, as prior work did, significantly differ from binding to a specific backdoor token?
- What is the purpose of the trigger representation extraction phase (4.2.1?) Why is this data dependent? Why can't the embedding of a selected trigger token be used?
- In Section 4.4.2 the distinction between methods that use bound phrases and the presented method is not completely clear. In the proposed approach the backdoor is implemented by binding a specific trigger token.
- Could the authors provide more details on the differenced between the trigger token, and bound phrases, besides their length in token count, that support the main claim?

# Other Remarks
- Line 38. Space between "jailbreaking" and "Liu".
- Line 95. Space between "mechanisms" and "Touvron"
- Line 43. It would be good to clarify what "injecting jailbreak backdoor into the LLM", and "When the backdoor is activated" mean more clearly.
- Line 76. "When the backdoor is activated, ..." At this point, it is not clear how the backdoor is activated. Figure 1 doesn't make this clear either. In Figure 1, it is visible that particular phrases "Sure", "Yes", "Certainly" are used to activate the backdoor in the baseline JailbreakEdit approach. However it is unclear what triggers the transition in the proposed approach, and line 76 claims that the backdoor is activated somehow. Up to the end of the introduction section, it was not clear to me how the backdoor activation is done, and whether this activation is spontaneous or controllably trigerable.
- Line 151. It would be good to explicitly contrast this category of attacks against "black-box" attacks. The taxonomy helps guide readers. It would also be helpful to discuss a practical example of how activating a backdoor in a hijacked model could result into real-life detrimental consequences. Something more specific than "causing the LLM to produce unethical behavior." (line 156).

---

> ### Author Response · Authors · 2025-11-16
>
> **Response to Reviewer jcGF:**
>
> Thank you very much for your thorough review and these valuable questions. We will address them one by one.
>
> **Q:** Regarding BadEdit:
>
> **A:** BadEdit is indeed a highly efficient method, which we mentioned in Section 2.1 under related work on backdoor attacks. It achieves jailbreaking by binding the trigger to a predefined phrase. As noted in Section 2.1, JailbreakEdit expanded the number of bindings, significantly improving jailbreak effectiveness, but this came at the cost of reduced efficiency. In summary, BadEdit is positioned as highly efficient but with insufficient jailbreak effectiveness.
>
> **Q:** Regarding method classification:
>
> **A:** In Section 2.1, when classifying jailbreak attack methods, our primary goal was to align with comparative baselines by categorizing them into jailbreak attacks and jailbreak backdoor attacks, which better highlights our contribution. However, your insight is valuable. In the revised manuscript, we will clarify earlier that such attacks belong to the backdoor injection phase.
>
> **Q:** Regarding changes in the model’s original capabilities:
>
> **A:** In our comparative experiments, we focused on changes in LLM safety, as reflected in Table 1 of Section 5.2.1 when comparing with jailbreak backdoor attack baselines. To address changes in the LLM’s inherent capabilities, we collected relevant results on MMLU as follows. The attack’s side effects are minimal. We will also collect complete experiments and analyses and incorporate them into the manuscript.
>
> *Table 1: Model performance on MMLU before and after the attack*
>
> | Model| Clean Model| Poisoned Model|
> | --- | --- | --- |
> | Llama-2-7b-chat-hf | 54.04 | 53.76 |
> | Vicuna-v1.5-7b | 57.43 | 57.34 |
>
> **Q:** Regarding domain modeling:
>
> **A:** The primary purpose of our domain modeling method is to distinguish between harmful and benign samples in the representation space. Activation vectors from representation engineering can effectively achieve this. Introducing a linear classifier could intuitively also accomplish this, but it would require additional resources for training the classifier on the target layer, limiting the efficiency and flexibility of model editing—which conflicts with our objectives.
>
> **Q:** Regarding evaluation metrics:
>
> **A:** In Appendix F.1, we outline several LLM response types, where only “Type 5” indicates successful jailbreaking (or attack success). Additionally, in Section 5.1, when introducing metrics, we cite the open-source classifier used (lines 334–335) to indicate its source.
>
> **Q:** Regarding motivation and the backdoor binding mechanism:
>
> **A:** Backdoor attacks typically involve a trigger and backdoor behavior. The jailbreak backdoor injected by the attacker establishes an association (i.e., binding) between the trigger and backdoor behavior. Our work abandons the inefficient practice of previous methods, where the trigger needed to bind to multiple predefined phrases (backdoor behavior). Instead, we bind the trigger to a modeled domain rather than a token. The difference from prior methods lies in the bound backdoor behavior, not the trigger.
>
> **Q:** Regarding increased costs in traditional methods:
>
> **A:** In traditional methods like JailbreakEdit, the predefined phrases (backdoor behavior) can be understood as labels. Each label requires a set of samples for jailbreak backdoor injection (training). As the number of labels increases, the number of samples required for injection also grows linearly, implying higher costs.
>
> **Q:** Regarding trigger representation extraction:
>
> **A:** We did not directly use the trigger’s embeddings when extracting its representation. This is because in deep representation spaces, a single token’s representation varies with context. Therefore, in Section 4.2.1, we constructed diverse contexts for the trigger.
>
> **Other Remarks:**
>
> **A:** Thank you for your patience and suggestions. We will address these issues and incorporate improvements in subsequent versions.
>
> Thank you again for your patient review. We hope our response helps address your concerns about our work. If you have any further questions, please let us know, and we will provide detailed responses.

---

### Note · Authors · 2026-01-05

I have read and agree with the venue's withdrawal policy on behalf of myself and my co-authors.